# MMDFND: Multi-modal Multi-Domain Fake News Detection

Yu Tong*
Wuhan University
Wuhan, China
yutchina02@gmail.com

Weihai Lu*
Peking University
Beijing, China
luweihai@pku.edu.cn

Zhe Zhao†
University of Science and Technology
of China
Hefei, China
City University of Hong Kong
Hong Kong, China
zz4543@mail.ustc.edu.cn

Song Lai
City University of Hong Kong
Hong Kong, China
songlai2-c@my.cityu.edu.hk

Tong Shi
Chinese University of Hong Kong,
Shenzhen
Shenzhen, China
224020346@link.cuhk.edu.cn

## ABSTRACT

Recently, automatic multi-domain fake news detection has attracted widespread attention. Many methods achieve domain adaptation by modeling domain category gate networks and domain-invariant features. However, existing multi-domain fake news detection faces three main challenges: (1) Inter-domain modal semantic deviation, where similar texts and images carry different meanings across various domains. (2) Inter-domain modal dependency deviation, where the dependence on different modalities varies across domains. (3) Inter-domain knowledge dependency deviation, where the reliance on cross-domain knowledge and domain-specific knowledge differs across domains. To address these issues, we propose a **M**ulti-modal **M**ulti-**D**omain **F**ake **N**ews **D**etection Model (MMDFND). MMDFND incorporates domain embeddings and attention mechanisms into a progressive hierarchical extraction network to achieve domain-adaptive domain-related knowledge extraction. Furthermore, MMDFND utilizes Stepwise Pivot Transformer networks and adaptive instance normalization to effectively utilize information from different modalities and domains. We validate the effectiveness of MMDFND through comprehensive comparative experiments on two real-world datasets and conduct ablation experiments to verify the effectiveness of each module, achieving state-of-the-art results on both datasets. The source code is available at https://github.com/yutchina/MMDFND.

## CCS CONCEPTS

• **Information systems** → **Social networks**; **Data mining**; • **Computing methodologies** → **Transfer learning**.

*Equal contribution.
†Corresponding author.

## KEYWORDS

Fake News Detection; Multi-domain; Multimodal Learning

**ACM Reference Format:**
Yu Tong, Weihai Lu, Zhe Zhao, Song Lai, and Tong Shi. 2024. MMDFND: Multi-modal Multi-Domain Fake News Detection. In *Proceedings of the 32nd ACM International Conference on Multimedia (MM'24), October 28-November 1, 2024, Melbourne, VIC, Australia* ACM, New York, NY, USA, 9 pages. https://doi.org/10.1145/3664647.3681317

## 1 INTRODUCTION

With the rapid advancement of communication technologies, individuals now have the ease of posting everyday information on social media and online platforms. However, certain users deliberately fabricate and disseminate false information to mislead the public, incite emotions, or achieve specific political or economic objectives, thereby causing severe consequences for societal stability. Moreover, distinguishing between true and false information often proves challenging for people, leading to the rapid spread of fake news. Manually verifying facts is not only costly but also time-consuming, hence, automatic Fake News Detection (FND) have garnered widespread attention [8, 45, 46].

Existing methods for fake news detection often focus solely on single-domain news, such as health or politics. However, news on social media in the real world frequently originates from multiple domains. This presents two challenges for single-domain fake detection methods: (1) When dealing with smaller, single-domain news datasets, the effectiveness of the model is significantly impacted due to the scarcity of training data; (2) The accuracy of single-domain fake news detection methods greatly diminishes on cross-domain news datasets. To address these challenges, some multi-domain fake news detection (MFND) methods have been proposed. Some previous works have employed a hard-sharing mechanism to learn domain-specific knowledge and cross-domain knowledge for multi-domain fake news detection [3, 31]. In addition, some methods based on a soft-sharing mechanism utilize domain-class gated networks to adjust the weights of multi-perspective information in text semantics, thereby achieving domain-adaptive multi-domain fake news detection [24, 47]. However, there are three problems with current multi-domain fake news detection methods:

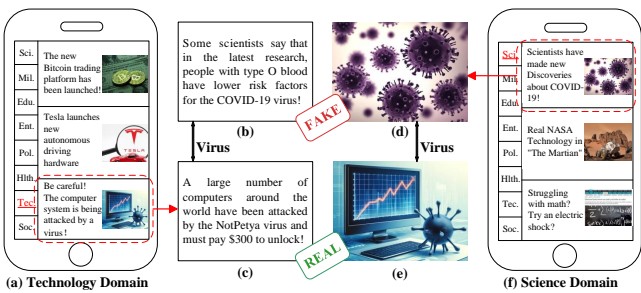

**Figure 1: An example of a multi domain news platform where there are certain differences in the semantics of news texts from different domains, as well as differences in the semantics of images from different domains.**

**(1) Inter-domain modal semantic deviation:** Texts and image descriptions from different domains exhibit varying degrees of semantic differences [2, 4, 10, 12, 20, 25, 33], which may lead to drastically different interpretations of the same news across different domains. For example, Figures 1(b) and 1(c) illustrate two news articles about "viruses," where in the "science" domain, "virus" typically refers to pathogens causing everyday diseases, while in the "technology" domain, it might refer to computer viruses. Additionally, Figures 1(d) and 1(e) display images from different news articles in two domains, where in the "science" domain, they often depict biological viruses, while in the "technology" domain, they may represent schematics of computer viruses. Even similar images may have different semantics in different domains. However, the existing MFND methods are all unimodal and have not taken into account the semantic bias between domains of multimodal information such as images and text.

**(2) Inter-domain modal dependency deviation:** Different domains vary in their reliance on modal information for decision-making, necessitating the consideration of multiple modalities [22, 35]. Some domains may prioritize the authenticity of text, while others may focus more on the authenticity of images, and certain scenarios may require the simultaneous consideration of both text and images for decision-making. For instance, the same artificially manipulated image may be classified as fake news in the "political" domain but as real news in the "entertainment" domain. Thus, different domains exhibit differences in the reliance on textual features, visual features, and multimodal features of the same news. However, existing MFND methods have not accounted for this dependency deviation during the modeling process, potentially leading to difficulties in achieving optimal performance.

**(3) Inter-domain knowledge dependency deviation:** Different domains exhibit varying degrees of reliance on cross-domain knowledge and domain-specific knowledge for decision-making. For example, domains with a higher density of specialized terminology may rely more on domain-specific knowledge, while others may depend more on cross-domain knowledge. However, existing MFND methods inadequately utilize domain-relevant knowledge by simply concatenating cross-domain knowledge and domain-specific knowledge, resulting in poor cross-domain predictive performance.

To address these challenges, we propose a multi-modal multi-domain fake news detection model called MMDFND. Initially, we extract news information from three distinct modal views. Specifically, we encode images using pre-trained CLIP [30] and MAE [13] encoders, while employing pre-trained CLIP text encoder and BERT [9] encoder for textual content. By fusing features from CLIP encodings through cross-modal similarity, we obtain multi-modal representations. To alleviate semantic biases across domains, we introduce the Domain Progressive Layered Extraction (DPLE) network for extracting cross-domain and domain-specific knowledge from different modalities. Subsequently, we fuse domain-relevant knowledge from different modal views using the Stepwise Pivot Transformer network to mitigate domain-modal dependency biases. Finally, we reweightedly aggregate domain-specific and cross-domain knowledge through Adaptive Instance Normalization. Through this approach, MMDFND effectively mitigates biases in domain knowledge dependencies, facilitating the effective utilization of domain-relevant knowledge across different modal views, thus enhancing the performance of multi-domain fake news detection. Our contributions are as follows:

- We propose MMDFND, a multi-modal multi-domain fake news detection model that comprehensively models information from different modalities across domains.
- We introduce domain embeddings and attention mechanisms within the Domain Progressive Layered Extraction network to achieve domain-adaptive extraction of domain-relevant knowledge.
- We employ the Stepwise Pivot Transformer network and adaptive instance normalization to proficiently harness information originating from diverse modalities and domains.
- We validate the effectiveness of MMDFND through extensive comparative experiments on two real-world datasets. Additionally, we demonstrate the effectiveness of each module within MMDFND through thorough ablation experiments and comparative analyses, achieving state-of-the-art results on both datasets.

## 2 RELATED WORKS

### 2.1 Unimodal Fake News Detection

Existing unimodal fake news detection technologies primarily depend on either single image information or single textual information. In terms of textual information analysis, [28] detects fake news by capturing linguistic information in the text of articles and generating new user responses. [1] utilizes TM's conjunction clauses to capture lexical and semantic properties of true and false news texts, achieving the detection of fake news. In addition to detecting fake news based on content features, text sentiment [43], writing style [26], and discourse-level structure [15] are also widely used in the detection of fake news. In the analysis of image information, MVNN [27] identifies fake news by integrating complex patterns of fake news images in the frequency domain with visual features in the pixel domain. However, these unimodal methods neglect cross-modal features and fail to utilize modal correlation information in the original data, such as the consistency and relevance between different modalities.

## 2.2 Multimodal Fake News Detection

Recently, some methods have utilized cross-modal discriminative patterns to detect fake news and have achieved outstanding performance [21, 40]. To learn a shared representation of multimodal information, [16] proposes a multimodal variational autoencoder, which can reconstruct a multimodal representation from the learned probabilistic latent models. CAFE [6] computes the Kullback-Leibler (KL) divergence between different unimodal features to measure cross-modal consistency. Before the final classification, the cross-modal consistency is used to adjust the weights of both unimodal and multimodal features. [37] guides the training of unimodal networks through the relevance of cross-modal features, then fuses the trained unimodal features for the detection of fake news. In this paper, we fuse features from different modal views using the Stepwise Pivot Transformer and adjust the fused weights adaptively, thereby effectively leveraging information from various modal views features.

## 2.3 Multi-Domain Fake News Detection

In the real world, news data often comes from diverse domains. The goal of multi-domain learning is to model data from multiple domains simultaneously, enhancing the performance of individual domains and thereby improving overall performance. Several approaches [7, 11, 19, 38] based on multi-domain learning have achieved remarkable results in the detection of fake news. MD-FEND [24] innovatively applies multi-domain learning to the detection of fake news. Specifically, it integrates representations extracted by multiple experts by feeding domain information into a gating network. To achieve multi-domain fake news detection, [47] extracts semantic, sentiment, and stylistic representations from textual information. It then adaptively aggregates the information of these three representations using a domain memory bank. However, these methods merely adjust the weights of different views or expert representations in text semantics based on domain embedding representations, failing to effectively learn and utilize domain-invariant and domain-specific information. EDDFN [31] learns features from domain-specific and cross-domain embeddings, then concatenates domain-relevant knowledge for fake news identification. However, this approach fails to consider the varying dependencies different domains have on cross-domain knowledge and domain-specific knowledge. Furthermore, all the above methods are single-modal approaches, which struggle to leverage the visual modality information present in news articles. In comparison, we extract cross-domain knowledge and domain-specific knowledge using domain-shared and domain-specific experts. We then aggregate domain-relevant representations of text, visual, and fused modalities through the pivot transformer, and reweight and aggregate cross-domain knowledge and domain-specific knowledge, effectively utilizing domain knowledge from multiple modal views.

## 3 PROBLEM STATEMENT

Each input of multimodal news is represented as $\mathcal{N} = [\mathbf{I}, \mathbf{T}] \in \mathcal{D}$, where $\mathbf{I}$, $\mathbf{T}$, and $\mathcal{D}$ respectively represent the image, text, and dataset. The news in the dataset is categorized into $k$ classes, each assigned a domain label $d \in \{Domain_1, \ldots, Domain_k\}$. Within the dataset, the quantity of news varies significantly across domains,

with some domains featuring a large number of articles, while others have relatively few. Given a news piece $\mathcal{N}$ that incorporates both text and image information, and a domain label $d$, the objective of multi-modal multi-domain fake news detection is to determine the authenticity of the news piece.

## 4 METHODOLOGY

In this section, we introduce our model, MMDFND, which achieves multi-domain fake news detection by extracting and fusing cross-domain and domain-specific knowledge from multiple modal views, and then reweighting these pieces of knowledge. The structure of the model is illustrated in Figure 2. Given an image-text pair, we first extract features from both single-modal and multi-modal views (§4.1). Subsequently, our approach comprises three crucial components: a module for extracting cross-domain and domain-specific knowledge (§4.2), a module for fusing multi-view representations (§4.3), and a module for reweighting the cross-domain and domain-specific knowledge (§4.4).

## 4.1 Multi-view Features Extraction

Employing pretrained models, we encode the image $\mathbf{I}$ and text $\mathbf{T}$ into single-modal embedding. We then merge the aligned single-modal embedding, obtained through the CLIP [30] encoder, to derive a multimodal feature representation.

*4.1.1 Visual View Feature.* Given a image $\mathbf{I}$, we employ the Masked Autoencoder (MAE) to extract relevant representations. These image representations are then converted into an image embedding, $e_i$, using fully connected layers. Furthermore, we obtain aligned image feature, $f_{\text{CLIP-I}}$, using the image encoder of CLIP.

*4.1.2 Text View Feature.* Given a text $\mathbf{T}$, We utilize a pre-trained BERT [9] model as the text encoder. By employing an MLP network, these text representations are transformed into text embeddings $e_t$. Furthermore, we leverage the text encoder from CLIP to obtain aligned text features $f_{\text{CLIP-T}}$.

*4.1.3 Multimodal View Feature.* Multimodal features reflect the correlations between two modes, while also encompassing a broader spectrum of semantic characteristics, enabling a more comprehensive analysis of the authenticity of news [42]. Therefore, we extract multimodal representations, considering it as a unique view. However, the image and text features extracted separately by MAE and BERT exhibit a significant cross-modal semantic gap, making their direct fusion difficult. To address this challenge, we employ the CLIP model, trained on a large-scale dataset of image-text pairs. By utilizing the CLIP model's encoder to encode both images and text, we obtain aligned image and text embeddings. These embeddings are then concatenated and fed into an MLP network to generate the multimodal representation. To address the ambiguity issues introduced by the classifier, we utilize the CLIP cosine similarity as a cross-modal similarity measure to weight the multimodal features. The resulting multimodal feature is denoted as $e_m$:

$$sim = \frac{f_{\text{CLIP-T}} \cdot (f_{\text{CLIP-I}})^T}{\|f_{\text{CLIP-T}}\| \|f_{\text{CLIP-I}}\|} \quad (1)$$

$$e_m = sim \cdot \text{MLP}(f_{\text{CLIP-I}} \oplus f_{\text{CLIP-T}}) \quad (2)$$

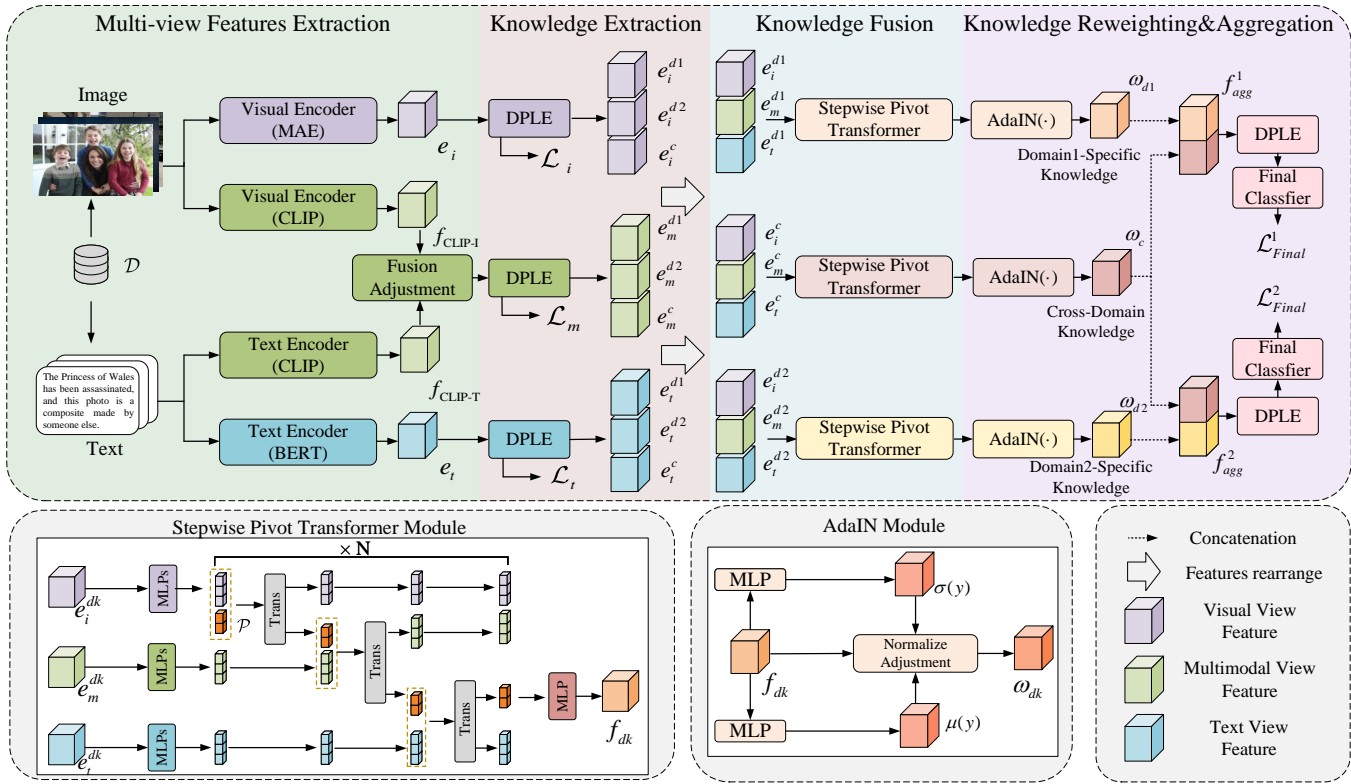

**Figure 2: The network architecture of MMDFND. BERT, MAE, and CLIP are utilized to encode the representations of different modal views of multi-modal news. Domain PLE extracts domain-specific knowledge and cross-domain knowledge from various modal views. The Stepwise Pivot Transformer fuses domain-relevant knowledge from different modal views. AdaIN adjusts the weights of cross-domain knowledge and domain-specific knowledge for different domains. The final decision on domain-specific fake news is based on the aggregated features of domain knowledge.**

## 4.2 Knowledge Extraction with Domain PLE

In this stage, our objective is to extract and learn both cross-domain knowledge and domain-specific knowledge from the representations obtained in the initial stage. To achieve this, we propose a Domain Progressive Layered Extraction (DPLE) model that enables the separate learning of cross-domain knowledge and domain-specific knowledge within the image, text, and multimodal representations.

As illustrated in Figure 3, the DPLE model comprises cross-domain experts, domain-specific experts, a gating network, domain embeddings, attention mechanism, and a classifier. The original PLE model assigns task-specific expert networks for each task. As depicted in Eq. (3), the PLE model extracts task-shared information and task-specific information from the data by sharing expert networks and task-specific expert networks, respectively, to model multi-task relationships. Specifically, the input $x$ is equally sent to both the shared expert network and the task-specific expert network. Subsequently, the gating network employs a SoftMax function to perform an adaptive weighted aggregation of both the shared and the task-specific expert networks, thus yielding the final representations for the different tasks. The variable $k$ denotes the number of distinct tasks.

$$x^k = \text{Softmax}(G^k(x))S^k(x) \tag{3}$$

where $G^k(x) \in R^{(m_s+m_k) \times d}$ represents the output of the gate network specific to the task $k$, and $d$ is the dimension of input representation. $S^k(x)$ represents the selection matrix consisting of chosen vectors, which includes $m_k$ task-specific experts for task $k$ and $m_s$ shared experts, $m_k$ and $m_s$ are two hyperparameters:

$$S^k(x) = \left[ E^T_{(k,1)}, E^T_{(k,2)}, \ldots, E^T_{(k,m_k)}, E^T_{(s,1)}, E^T_{(s,2)}, \ldots, E^T_{(s,m_s)} \right]^T \tag{4}$$

In MMDFND, we treat fake news detection in different domains as distinct tasks. In this paper, we take two domains as examples. We leverage DPLE to unearth both cross-domain and domain-specific knowledge. We enhance the PLE network from two perspectives. Firstly, we compute the importance scores for each token using an MLP network with shared weights based on an attention mechanism, and perform aggregation according to these scores. Subsequently, we feed the aggregated low-dimensional representation into a gating network as part of its input. Secondly, acknowledging that domain experts excel more in specific fields than cross-domain experts, we aim to select appropriate experts via the gating network. Therefore, we incorporate a learnable feature $e^d$, namely domain embedding, as another input to the gating network, thereby guiding the selection process of the gating network. We revise Eq. (3) as Eq.

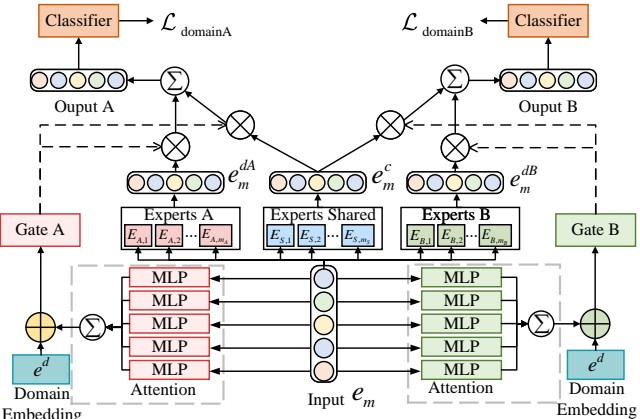

**Figure 3: The network architecture of Domain PLE**

(5) for DPLE:

$$x^k = \text{Softmax}(G^k(\sum_{i=1}^{t} \text{MLP}_k(x) \oplus e^d))S^k(x) \tag{5}$$

where $t$ represents the number of tokens, and $\text{MLP}_k$ denotes the attention network used for domain $k$. $k \in [1, 2]$. The classifier consists of an MLP network and a Softmax output layer. By inputting the results of the two gating networks into their corresponding classifiers, the predicted values for the two domains can be obtained:

$$\hat{y}_k = \text{Softmax}(\text{MLP}(x^k)) \tag{6}$$

The purpose of fake news detection is to determine the veracity of news articles. We use $\hat{y}_k$ to represent the predicted label in domain $k$ and $y_k$ to represent the true value in domain $k$. We employ the Binary Cross-Entropy (BCE) loss as the loss function in domain $k$:

$$\mathcal{L}_{\text{domaink}} = -(y_k \log \hat{y}_k + (1 - y_k) \log(1 - \hat{y}_k)) \tag{7}$$

By calculating the weighted sum of the losses across different domains, we obtain the joint loss for multi-domain learning:

$$\mathcal{L}_{DPLE} = \sum_{k=1}^{K} \omega_k \mathcal{L}_{\text{domaink}} \tag{8}$$

Where $K$ is the number of domains. Constrained by the loss function of the DPLE network, we utilize the cross-domain expert network and domain-specific expert networks to extract cross-domain knowledge of visual modality, denoted as $e_i^c$, and domain-specific knowledge for two domains, represented as $[e_i^{d1}, e_i^{d2}]$, from $e_i$. From $e_t$, we extract cross-domain knowledge of textual modality, denoted as $e_t^c$, and domain-specific knowledge for the same two domains, represented as $[e_t^{d1}, e_t^{d2}]$. From $e_m$, we extract cross-domain knowledge of multimodal, denoted as $e_m^c$, and domain-specific knowledge for the same two domains, represented as $[e_m^{d1}, e_m^{d2}]$. To effectively utilize domain-specific knowledge and domain-shared knowledge from different modalities, domain-shared knowledge and domain-specific knowledge from the same domain will be fused in subsequent steps through a pivotal Transformer.

## 4.3 Multi-view Knowledge Fusion with Stepwise Pivot Transformer

In the knowledge fusion stage, our goal is to integrate domain-specific knowledge and cross-domain knowledge from different views. To achieve this, we draw upon ideas from MMSBR [44] and propose the Stepwise Pivot Transformer.

*4.3.1 Feature Sequence Generation.* We generate feature sequences from different views using Feed-Forward Networks (FFN). Taking cross-domain knowledge from text ($e_t^c$), image ($e_i^c$), and multimodal ($e_m^c$) views as examples, the feature sequences from different views can be represented as:

$$S_t^c = \{FN_1^t(e_t^c), \ldots, FN_R^t(e_t^c)\} \tag{9}$$

$$S_i^c = \{FN_1^i(e_i^c), \ldots, FN_R^i(e_i^c)\} \tag{10}$$

$$S_m^c = \{FN_1^m(e_m^c), \ldots, FN_R^m(e_m^c)\} \tag{11}$$

where $S_t^c$, $S_i^c$ and $S_m^c$ represent the cross-domain knowledge feature sequences of text, image, and multimodal views, respectively, and $R$ denotes the length of the sequence.

*4.3.2 Stepwise Pivot Transformer.* A standard Transformer layer primarily consists of three components: Multi-Head Attention (MHA), Layer Normalization (LN), and Feed-Forward Networks (FFN). Therefore, the Transformer layer as $F^{l+1} = \text{Trans}(F^l)$ can be represented as follows, where $F^l$ is the input sequence, and $F^{l+1}$ represents the output sequence:

$$F^l = \text{MHA}(\text{LN}(F^l)) + F^l \tag{12}$$

$$F^{l+1} = \text{FFN}(\text{LN}(F^l)) + F^l \tag{13}$$

In the initial Transformer layer, we create a pivot $\mathcal{P} = [p_1, \ldots, p_M]$ to gradually fuse information from different views. Taking the fusion of cross-domain knowledge from various views as an example, the Stepwise Pivot Transformer integrates information from visual ($S_i^c$), multimodal ($S_m^c$) and textual ($S_t^c$) sequences through the following process:

$$[S_i^{c,l+1}, \mathcal{P}_i^l] = \text{Trans}([S_i^{c,l}, \mathcal{P}^l]) \tag{14}$$

$$\mathcal{P}^l = (\mathcal{P}_i^l + \mathcal{P}^l)/2 \tag{15}$$

$$[S_m^{c,l+1}, \mathcal{P}_m^l] = \text{Trans}([S_m^{c,l}, \mathcal{P}^l]) \tag{16}$$

$$\mathcal{P}^l = (\mathcal{P}_m^l + \mathcal{P}^l)/2 \tag{17}$$

$$[S_t^{c,l+1}, \mathcal{P}_t^l] = \text{Trans}([S_t^{c,l}, \mathcal{P}^l]) \tag{18}$$

$$\mathcal{P}^l = (\mathcal{P}_t^l + \mathcal{P}^l)/2 \tag{19}$$

where $S_i^{c,0} = S_i$, $S_m^{c,0} = S_m$, $S_t^{c,0} = S_t$ and $\mathcal{P}^0 = \mathcal{P} = (S_t^{c,0} + S_i^{c,0} + S_m^{c,0})/3$, In the above process, the pivot integrates information from different views. For instance, as illustrated by Eq. (16), the pivot, which has merged textual information, provides insights from the textual view to the image view, thereby fusing information from both text and image views through the Transformer layer. To fully integrate information from different views, we stacked the Stepwise Pivot Transformer $N$ times. After the final fusion, the information encapsulated within the pivot $\mathcal{P}^N$ is extracted through an MLP network, denoted as $f_c$:

$$f_c = \text{MLP}(\mathcal{P}^N) = \text{MLP}(p_1^N \oplus p_2^N, \ldots, \oplus p_M^N) \tag{20}$$

Through Stepwise Pivot Transformer, we can also obtain domain-specific information $f_{dk}$ for domain $k$ after multi-view fusion. $k \in [1, 2]$. These domain related knowledge will be re weighted and aggregated in subsequent stages.

## 4.4 Knowledge Reweighting

After the process of knowledge fusion, we have obtained domain-specific knowledge and cross-domain knowledge that incorporate multi-view information. To effectively utilize this knowledge, in this stage, we adaptively reweight and aggregate domain-specific knowledge and cross-domain knowledge.

We leverage Adaptive Instance Normalization (AdaIN) method for reweighting. By training the mean and variance of the original features through an MLP, and then using the trained mean and variance to guide the adjustment of the normalized feature representation, we can selectively enhance the knowledge that is more important to the current domain while reducing the impact of knowledge that contributes less. Taking the generation of the reweight feature $\omega_c$ based on $f_c$ as an example, the specific formula is as follows:

$$\sigma(y) = \text{MLP}(f_c), \mu(y) = \text{MLP}(f_c) \tag{21}$$

$$\omega_c = \sigma(y) \left( \frac{f_c - \mu(f_c)}{\sigma(f_c)} \right) + \mu(y) \tag{22}$$

Similarly, we generate the reweighted representation for the domain-specific knowledge of domain $k$, denoted as $\omega_{dk}$. $k \in [1, 2]$.

## 4.5 Aggregation Stage and Loss Function

The reweighted representations $[\omega_{dk}, \omega_c]$ are concatenated and fed into the DPLE, resulting in a feature $f_{agg}^k$ that is enriched with both domain-specific and cross-domain knowledge:

$$f_{agg}^k = \text{DPLE}(\omega_{dk} \oplus \omega_c) \tag{23}$$

$f_{agg}^k$ is subsequently fed into the final classifier, yielding the ultimate prediction of news veracity, denoted as $\hat{y_k}$. Given that fake news detection constitutes a binary classification challenge, we employ a Binary Cross-Entropy (BCE) loss for the final classifier, represented as $\mathcal{L}_{Final}^k$. Moreover, according to Eq. (8), the losses for the text view, visual view, and fusion view branches within the DPLE model are represented as $\mathcal{L}_t$, $\mathcal{L}_i$, and $\mathcal{L}_m$, respectively. The total loss for the DPLE model classification is the sum of $\mathcal{L}_t$, $\mathcal{L}_i$, and $\mathcal{L}_m$, indicated as $\mathcal{L}_{DPLE}$. Hence, the overall loss for domain $k$ in the MMDFND model is presented as follows:

$$\mathcal{L}^k = \mathcal{L}_{Final}^k + \alpha \mathcal{L}_{DPLE} = \mathcal{L}_{Final}^k + \alpha(\mathcal{L}_t + \mathcal{L}_i + \mathcal{L}_m) \tag{24}$$

The DPLE loss is invariant across domains. Therefore, the total loss of the MMDFND model is computed as the sum of the losses from the final classifiers across different domains, augmented by the DPLE loss. The loss function for the MMDFND model is articulated as follows:

$$\mathcal{L} = \sum_{k=1}^{K} \mathcal{L}_{Final}^k + \alpha \mathcal{L}_{DPLE} \tag{25}$$

Where $K$ denotes the number of domains, while $\alpha$ represents hyperparameter.

## 5 EXPERIMENTS

In this section, we empirically evaluate MMDFND through two benchmark tests covering news data from different domains, aiming to address the following research questions:

- **RQ1**:Can our proposed MMDFND outperform the latest baselines in multi-domain fake news detection and multi-modal fake news detection?
- **RQ2**:What is the impact of each component in MMDFND?
- **RQ3**: Can MMDFND improve the performance of fake news detection in data-scarce domains?

## 5.1 Experimental Settings

*5.1.1 Datasets.* Our model is evaluated on two real-world datasets: Weibo [36] and Weibo-21 [24]. For the Weibo dataset, we adhere to the same data partitioning method as the benchmark [36, 41]. The training set consists of 3,749 real news items and 3,783 fake news items, while the testing set comprises 1,000 fake news items and 996 real news items. To meet the requirements of multi-domain datasets, we divided the Weibo dataset into nine domains, namely, finance, health, military, science, politics, international, education, entertainment, and society. Weibo-21, a new multi-domain and multi-modal dataset, contains a total of 4,640 real news items and 4,487 fake news items; it is divided into training and testing data following the partitioning approach of benchmark [47]. Besides, to ensure the quality of the entire dataset, we follow the same steps as in the work [14, 39] to remove duplicated and low-quality images.

*5.1.2 Baseline.* To conduct a comprehensive evaluation of our proposed model, we compare it with both **multi-domain (MD)** fake news detection methods and **multimodal (MM)** fake news detection approaches.

**MD1: MMoE** [23], which is a multi-domain model that shares a mixture of experts (MoE) across various domains.

**MD2: MoSE** [29], which employs Long Short-Term Memory networks (LSTM) as the experts within the MMoE framework.

**MD3: EDDFN** [31], which retains domain-specific and domain-shared knowledge extracted from text and propagation information for multi-domain fake news detection.

**MD4: MDFEND** [24], which is a multi-domain fake news detection model that utilizes a domain gate to perform weighted aggregation of MoE experts.

**MD5: M³DFEND** [47], which replaces the experts in MDFEND with text semantic, sentiment, and style extractors, while substituting domain gating with domain adapters.

**MM1: EANN** [36], which is a model grounded in Generative Adversarial Networks (GAN) capable of learning event-invariant representations.

**MM2: SpotFake** [32], which utilizes VGG and BERT to extract image and text features, respectively, for fake news detection.

**MM3: CAFE** [6], which utilizes cross-modal ambiguity to adaptively aggregate unimodal features and cross-modal correlations.

**MM4: CMC** [37], which introduces a novel distillation method to extract cross-modal correlations during training.

**MM5: BMR** [41], which aggregates multi-view features and cross-modal consistency through weighted integration.

**Table 1: Comparison between MMDFND and the latest multi-domain fake news detection methods on Weibo and Weibo-21. *: open-source.**

| Datasets | Method | Sci. | Mil. | Edu. | Soc. | Pol. | Hlth. | Fin. | Ent. | Dis./Int | overall | | |
|---|---|---|---|---|---|---|---|---|---|---|---|---|---|
| | | | | | | | | | | | F1 | Acc | Auc |
| Weibo | MMoE* | 0.578 | 0.911 | 0.851 | 0.885 | 0.735 | 0.826 | 0.813 | 0.830 | 0.883 | 0.874 | 0.874 | 0.950 |
| | MoSE* | 0.793 | 0.738 | 0.834 | 0.912 | 0.764 | 0.859 | 0.791 | 0.844 | 0.883 | 0.890 | 0.890 | 0.954 |
| | EDDFN | 0.566 | 0.823 | 0.838 | 0.848 | 0.735 | 0.851 | 0.754 | 0.802 | 0.887 | 0.855 | 0.855 | 0.934 |
| | MDFEND* | 0.774 | 0.911 | 0.897 | 0.902 | 0.763 | 0.878 | 0.808 | 0.881 | 0.874 | 0.904 | 0.904 | 0.965 |
| | M$^3$FEND | 0.792 | 0.903 | 0.923 | 0.912 | **0.765** | 0.863 | 0.899 | 0.899 | 0.876 | 0.928 | 0.928 | 0.969 |
| | **MMDFND** | **0.824** | **0.911** | **0.941** | **0.939** | 0.735 | **0.913** | **0.917** | **0.917** | **0.888** | **0.934** | **0.934** | **0.972** |
| Weibo-21 | MMoE* | 0.875 | 0.911 | 0.870 | 0.875 | 0.862 | 0.936 | 0.856 | 0.888 | 0.877 | 0.894 | 0.894 | 0.954 |
| | MoSE* | 0.850 | 0.885 | 0.881 | 0.872 | 0.880 | 0.917 | 0.8672 | 0.891 | 0.867 | 0.893 | 0.894 | 0.954 |
| | EDDFN* | 0.818 | 0.913 | 0.867 | 0.868 | 0.847 | 0.937 | 0.863 | 0.883 | 0.878 | 0.891 | 0.891 | 0.952 |
| | MDFEND* | 0.830 | 0.938 | 0.891 | 0.898 | 0.886 | 0.940 | 0.895 | 0.906 | 0.900 | 0.913 | 0.913 | 0.970 |
| | M$^3$FEND* | 0.829 | 0.950 | **0.899** | 0.908 | 0.882 | **0.946** | **0.900** | 0.931 | 0.889 | 0.921 | 0.921 | 0.975 |
| | **MMDFND** | **0.937** | **0.953** | 0.852 | **0.945** | **0.965** | 0.920 | 0.884 | **0.959** | **0.919** | **0.939** | **0.939** | **0.977** |

*5.1.3 Implementation Details.* In the text encoding section, we set the maximum input length for text at 197 characters. The pretrained BERT [9] and CLIP models are utilized to encode the text data respectively. For the visual encoding portion, the input images are resized to 224x224 pixels, and encoded separately using pretrained MAE [13] and CLIP models. We utilize TextCNN [5] as the expert for the textual view of DPLE, CNN [18] as the expert for the visual view of DPLE, and MLP as the expert for the multimodal view of DPLE. Our evaluation metrics encompass Accuracy, Precision, Recall, and the F1 Score. The experiments are conducted with a batch size of 64, employing the Adam optimizer [17] with an initial learning rate of 0.0001. The models are trained over 50 epochs with early stopping implemented to prevent overfitting. In the DPLE loss equation (Eq. 8), equal weight is assigned across different domains, whereas in the MMDFND model loss equation (Eq. 25), the parameters $\alpha$ is set to 0.15. We set the number of both domain-specific experts $m_k$ and shared experts $m_s$ to 6.

## 5.2 Overall Performance (RQ1)

To assess our proposed method from both domain-specific and overall perspectives, we conduct comparisons between MMDFND and various cutting-edge multi-domain and multimodal fake news detection methodologies on the Weibo and Weibo21 datasets, respectively. As indicated in Table 1, MMDFND notably surpasses other multi-domain strategies in F1 scores across several domains and overall. Demonstrated in Table 2, MMDFND significantly outperforms alternative multimodal approaches in terms of accuracy and F1 scores across each dataset, substantiating the efficacy of our proposed model. Specifically, MMDFND achieves unprecedented accuracy rates of **93.4%** on the Weibo dataset and **93.9%** on the Weibo21 dataset, achieving a new state-of-the-art on Weibo and Weibo21 datasets.The results lead to the following conclusions:

In multi-domain approaches, MDFEND and M$^3$FEND outperform MMoE, MoSE, and EDDFN across the majority of domains. MMoE and MoSE employ a shared expert base with multiple independent heads for different domains. EDDFN learns domain-specific and cross-domain knowledge separately. These three methodologies all adopt hard sharing mechanisms to acquire cross-domain

**Table 2: Comparison between MMDFND and the latest multi-modal fake news detection methods on Weibo and Weibo-21. *: open-source.**

| Datasets | Method | Accuracy | F1 score | |
|---|---|---|---|---|
| | | | Fake News | Real News |
| Weibo | EANN* | 0.827 | 0.829 | 0.825 |
| | SpotFake* | 0.892 | 0.932 | 0.739 |
| | CAFE* | 0.840 | 0.842 | 0.837 |
| | CMC | 0.893 | 0.899 | 0.907 |
| | BMR* | 0.918 | 0.914 | 0.904 |
| | **MMDFND** | **0.934** | **0.936** | **0.932** |
| Weibo-21 | EANN* | 0.870 | 0.862 | 0.875 |
| | SpotFake* | 0.851 | 0.828 | 0.866 |
| | CAFE* | 0.882 | 0.885 | 0.876 |
| | CMC* | 0.897 | 0.903 | 0.912 |
| | BMR* | 0.929 | 0.927 | 0.925 |
| | **MMDFND** | **0.939** | **0.940** | **0.939** |

knowledge. However, these methods are unable to effectively aggregate cross domain knowledge and domain specific knowledge. Conversely, MDFEND and M$^3$FEND leverage soft sharing mechanisms to aggregate cross-domain knowledge, resulting in superior performance against all baseline methods. Nevertheless, unimodal detection methods, limited by the sparse information they collect and relying solely on soft sharing mechanisms without employing hard sharing approaches, fail to harness cross-domain knowledge effectively, leading to suboptimal performance.

In multimodal approaches, EANN's neglect of semantic differences leads to poor fusion and the worst performance. SpotFake excels in rumor classification using pre-trained models for separate text and image features but performs weakly in non-rumor classification. CAFE improves multimodal fusion by aligning semantic spaces through auxiliary tasks, achieving better performance. CMC and BMR show strong overall performance by recognizing cross-modal correlations but fall short on cross-domain datasets due to their lack of cross-domain and domain-specific knowledge.

**Table 3: Ablation study on the network design of MMDFND on two datasets.**

| Dataset | Method | Accuracy | F1 score | |
|---------|--------|----------|-----------|-----------|
| | | | Fake News | Real News |
| Weibo | **MMDFND** | **0.934** | **0.936** | **0.932** |
| | -w/o MVF | 0.911 | 0.917 | 0.904 |
| | -w/o ITA | 0.893 | 0.899 | 0.886 |
| | -w/o DPLE | 0.904 | 0.913 | 0.895 |
| | -w/o ReW | 0.922 | 0.929 | 0.914 |
| | -w/o Cross | 0.878 | 0.887 | 0.868 |
| Weibo-21 | **MMDFND** | **0.939** | **0.940** | **0.939** |
| | -w/o MVF | 0.923 | 0.925 | 0.921 |
| | -w/o ITA | 0.917 | 0.917 | 0.917 |
| | -w/o DPLE | 0.916 | 0.919 | 0.914 |
| | -w/o ReW | 0.926 | 0.927 | 0.925 |
| | -w/o Cross | 0.910 | 0.910 | 0.910 |

The superiority of MMDFND over other state-of-the-art methods in multimodal and multi-domain datasets can be attributed primarily to the following reasons: (1) MMDFND combines hard and soft sharing mechanisms, using hard sharing to extract cross-domain and domain-specific knowledge, and soft sharing to aggregate and weigh this knowledge, enhancing predictive performance across multiple domains. (2) From an information-theoretic perspective, MMDFND uses cross-domain knowledge to enhance data-scarce domains, improving the predictive performance of these single domains and overall performance. (3) MMDFND aligns semantic spaces across modalities using CLIP's encoder, enhancing multimodal fusion. (4) MMDFND utilizes the Stepwise Pivot Transformer to integrate information from different modal views, enabling fake news detection from multiple angles.

## 5.3 Ablation Studies (RQ2)

To evaluate the effectiveness of each component in the MMDFND framework, we perform comparative analyses by omitting each component individually. The experimental setups are as follows: **(1) w/o MVF:** The multi-view knowledge fusion module is omitted, and fusion is performed by concatenating features from different views. **(2) w/o ITA:** The CLIP encoder, which aligns the graphical and textual modalities, is removed. Encoding of different modalities is performed exclusively using the BERT and MAE models. **(3) w/o DPLE:** The enhancements to the PLE model are removed, relying solely on the original PLE model to extract domain-specific and cross-domain knowledge. **(4) w/o ReW:** The reweighting mechanism for aggregating cross-domain and domain-specific knowledge is eliminated. **(5) w/o Cross:** The cross-domain knowledge that supplements information from other domains is omitted.

Table 3 presents the results of ablation studies. We observe that the original MMDFND outperforms all variants, demonstrating the efficacy of each component. Moreover, our findings lead to the following conclusions:

- MMDFND performs poorly without cross-domain knowledge, highlighting its crucial role in enhancing multi-domain learning.

- The performance of MMDFND with the original PLE model drops significantly, proving our enhancements improve cross-domain and domain-specific knowledge extraction.

## 5.4 Visualization Results (RQ3)

To verify if our model improves fake news detection in data-scarce domains, we used t-SNE [34] to visualize features before each domain classifier (Figure 4). The results show that MMDFND excels even in domains with limited data, such as international, finance, and science. This is because MMDFND transfers knowledge from data-rich domains (like health, entertainment, and social) to enhance detection in data-scarce areas. Additionally, MMDFND adaptively reweights these knowledge sources, combining cross-domain and domain-specific insights to boost overall performance.

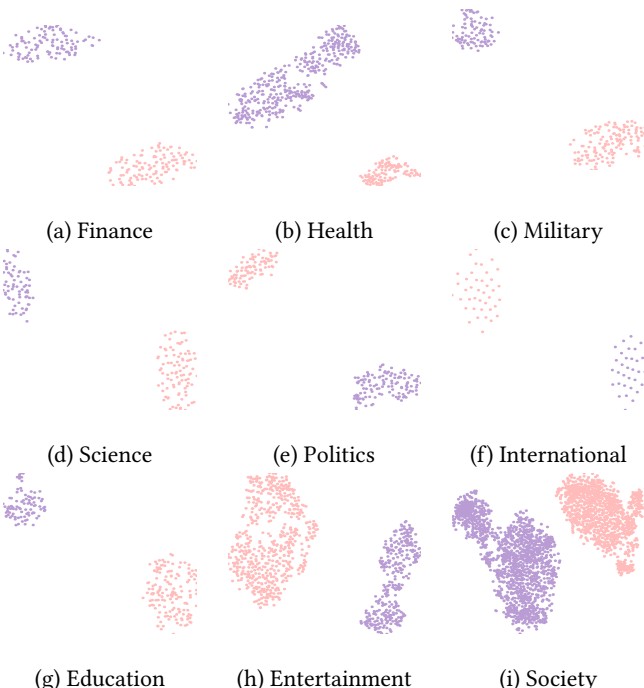

(a) Finance        (b) Health        (c) Military

(d) Science        (e) Politics        (f) International

(g) Education        (h) Entertainment        (i) Society

**Figure 4: T-SNE visualizations display the prediction results of MMDFND across different domains on the Weibo test dataset, where dots of the same color indicate the same label.**

## 6 CONCLUSION

This paper proposes MMDFND, a multi-modal, multi-domain fake news detection model. It incorporates domain embeddings and attention mechanisms into a progressive hierarchical extraction network to achieve domain-adaptive domain-related knowledge extraction. Besides, MMDFND utilizes Stepwise Pivot Transformer networks and adaptive instance normalization to effectively utilize information from different modalities and domains. We validate the effectiveness of MMDFND through comprehensive comparative experiments on two real-world datasets and conduct ablation experiments to verify the effectiveness of each module, achieving state-of-the-art results on both datasets.

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
