# OpenReview forum: "MMDFND: Multi-modal Multi-Domain Fake News Detection"
_acmmm.org/ACMMM/2024/Conference — MM2024 Poster_

### Official Review · Reviewer_kRkA · 2024-05-23

**Rating:** 3
**Confidence:** 3

**Summary:**

This paper proposed MMDFND to solve the problem of multi-modal multi-domain fake news detection, which encodes images and text via different pre-trained models to get single-modal features and fused multi-modal features. Domain Progressive Layered Extraction modules are further exploited to extract cross-domain and domain-specific features for different model views. They exploit Stepwise Pivot Transformers to fuse multi-modal knowledge for cross-domain and domain-specific branches for further aggregation. Experimental results on Weibo datasets demonstrate the effectiveness of MMDFND.

**Strengths:**

The research question is important, and this paper proposed an effective method to solve multi-modal multi-domain fake news detection problem.

**Limitations:**

(1) As far as I know, **Knowledge augmented transformer for adversarial multidomain multiclassification multimodal fake news detection,
Neurocomputing** is an earlier research that focuses on multi-modal multi-domain fake news detection but missed in your related work and compared methods.

(2) I think the factuality of one image is independent of the domain, so can you provide some examples to prove **the same artificially manipulated image may be classified as fake news in the "political" domain but as real news in the "entertainment" domain.** mentioned in lines 106-108?

(3) I think the formula (5) is incorrect because $x$ does not represent a token but the input to one MLP module is a token of $x$ according to Figure 3.

(4) In Section 4.5, this paper describes that the aggregated feature $f^k_{agg}$ is acquired by concatenating $\omega_{dk}$ and $\omega_{c}$ and feeding into the DPLE, which is not consistent with the Formula (23).
Moreover, I am confused about the reason that you also use the DPLE in the aggregation stage, in which domain-specific and cross-domain knowledge is prepared well. How about just fusing them via concatenating or simple weighting？

**Suitability:**

2

---

### Official Review · Reviewer_D1qW · 2024-05-25

**Rating:** 5
**Confidence:** 3

**Summary:**

The paper presents a novel approach to detecting fake news across multiple domains and modalities, titled "MMDFND: Multi-modal Multi-Domain Fake News Detection." It addresses key challenges in fake news detection that arise due to variations in semantics, modal dependencies, and knowledge dependencies across different domains. The proposed model, MMDFND, integrates multi-modality (text and images) and multi-domain (various content domains) considerations into a single framework.

Key features of MMDFND include the use of domain embeddings and attention mechanisms to adaptively extract domain-specific knowledge, and the incorporation of Stepwise Pivot Transformer networks and adaptive instance normalization to handle information from various modalities effectively.

The effectiveness of the MMDFND model is validated through comprehensive experiments on two real-world datasets, demonstrating its ability to outperform existing methods in accuracy and adaptability across different domains. The paper also includes detailed ablation studies to verify the importance of each component in the model.

**Strengths:**

1.The paper introduces the MMDFND model, an innovative approach that uniquely combines multi-domain and multi-modal data, enabling it to tackle the complex and varied nature of fake news effectively across different content types and formats.

2. The model is robustly validated with extensive experiments on real-world datasets and detailed ablation studies. This comprehensive testing demonstrates the model’s effectiveness and the critical role of each component, enhancing the paper's credibility.

3. MMDFND achieves state-of-the-art performance, outperforming existing methods in fake news detection. This superior performance across diverse domains and modalities illustrates the model's potential to significantly impact the fight against misinformation.

**Limitations:**

1. The manuscript introduces the Stepwise Pivot Transformer, which gradually integrates different types of information (textual, visual, multimodal) through a series of Transformer layers, detailing the process extensively. However, there may still be issues of information loss or insufficiency during each step of the information updating process. Information from different views may present conflicts or inconsistencies, posing a challenge on how to effectively address these conflicts.

2. Regarding the hyper parameter α in the formula, please provide some details about its selection and tuning. For example, how was the initial value of α determined? Was cross-validation performed to adjust the value of α? Such information helps readers understand the process and decisions involved in model training.

3. In the experimental setup, some details may require further explanation, such as why specific input lengths were chosen, why certain optimizers and learning rates were used, why specific parameters were set in the model's loss function, and so on.

**Suitability:**

3

---

### Official Review · Reviewer_Wc2d · 2024-05-25

**Rating:** 4
**Confidence:** 3

**Summary:**

The paper titled "MMDFND: Multi-modal Multi-Domain Fake News Detection" addresses the challenges in automatic multi-domain fake news detection. The authors propose a novel model, MMDFND, which is the first to jointly model multi-modality and multi-domain in the fake news detection scenario.

**Strengths:**

1. The use of domain embeddings and attention mechanisms in a progressive hierarchical extraction network seems innovative.
2. The authors conduct comprehensive comparative experiments on two real-world datasets, demonstrating the model's effectiveness and achieving state-of-the-art results.

**Limitations:**

1.	Ambiguity in Terminology: The use of "another" in line 144 is ambiguous and could be clarified.
2.	Unused Parameters: Some parameters are introduced but not used in the text, formulas, or figures, such as “The hyperparameter 𝑛” in line 346.
3.	Inconsistent Abbreviations: The abbreviation for "Feed-forward Networks" is inconsistent, with "(FN)" used in line 512 and "(FFN)" in line 528.
4.	Inconsistent Formula and Figure Order: The order of formulas (14)-(19) does not match the sequence shown in Figure 1 for the "Stepwise Pivot Transformer Module".

**Suitability:**

2

---

### Official Review · Reviewer_QuY5 · 2024-05-27

**Rating:** 4
**Confidence:** 2

**Summary:**

The paper is about fake news detection, which is an interesting topic. The authors propose a Multi-modal Multi-Domain Fake News Detection Model (MMDFND), the first to jointly model multi-modality and multi-domain in the fake news detection scenario. However, there are several concerns in the current version of the paper that addressing them will increase the quality of this paper.

**Strengths:**

[1] Novel ideas and research questions.

[2] Reasonable writing logic.

[3] sufficient experimental results.

**Limitations:**

1 Multi-modal scenarios should be relatively common in fake news detection. The author claims that this is the first work. Is the main focus on multi-domain? In this regard, the author should provide further clarification.

2 Figure 2 can be further improved. At present, a clear model structure can be seen, but there needs to be a clear logic/process line.

3 Is it not detailed enough to use the same hyperparameter to adjust the weights of different loss functions? Have you considered using multiple hyperparameters to correspond to different loss functions?

4 Using graph technology to model multi-modal data seems to be an optional path, which is also common in fake news detection. It is recommended that the authors increase the discussion on this part. [1-3]
[1] Improving fake news detection by using an entity-enhanced framework to fuse diverse multimodal clues. 2021.
[2] TMac: Temporal multi-modal graph learning for acoustic event classification. ACM MM 2023.
[3] Multi-modal graph contrastive learning for micro-video recommendation. ACM MM 2022.

**Suitability:**

3

---

### Official Review · Reviewer_DAa6 · 2024-05-27

**Rating:** 4
**Confidence:** 2

**Summary:**

MMDFND sets a new benchmark in the field of fake news detection by effectively combining multi-modal and multi-domain information through advanced mechanisms like domain embeddings, attention, Stepwise Pivot Transformer networks, and adaptive instance normalization. The extensive experimental validation underscores its robustness and superior performance.

**Strengths:**

1. The authors consider that existing methods  faces three main challenges: (1) Inter-domain modal semantic deviation, (2) Inter-domain modal dependency deviation, and (3) Inter-domain knowledge dependency deviation.
2. The authors propose MMDFND, a multi-modal multi-domain fake news detection model that comprehensively models information from different modalities across domains. MMDFND stands as the first model to address cross-domain fake news detection leveraging multi-modal domain-relevant information.
3. The proposed network achieves state-of-the-art performance on two public real-world datasets.

**Limitations:**

1. The author's claim that this is the first to jointly model multi-modality and multi-domain in the fake news detection scenario is inaccurate[1,2].
2. The proposed network achieves state-of-the-art performance. But one thing that seems missing but potentially important is the efficiency (e.g., number of parameters in the model, number of examples that can be processed per second and time-sensitivity) of the system.
3. The related work section lacks an introduction to recent studies from the past one or two years. For example, the section on Unimodal Fake News Detection is missing research from 2022 and 2023, and the Multimodal Fake News Detection section is missing research from 2023.
4. The selection of parameters α, m_k, and m_s needs to be supported by experimental results, so the author can add related parameter analysis.
5. Please check if formulas 14 to 19 are written incorrectly.


[1] Silva A, Luo L, Karunasekera S, et al. Embracing domain differences in fake news: Cross-domain fake news detection using multi-modal data[C]//Proceedings of the AAAI conference on artificial intelligence. 2021, 35(1): 557-565.
[2] Zhang T, Wang D, Chen H, et al. BDANN: BERT-based domain adaptation neural network for multi-modal fake news detection[C]//2020 international joint conference on neural networks (IJCNN). IEEE, 2020: 1-8.

**Suitability:**

3

---

### Official Review · Reviewer_NkeJ · 2024-05-30

**Rating:** 4
**Confidence:** 3

**Summary:**

In this study, a multi-modality and multi-domain fake news detection method MMDFND is proposed to incorporate domain embeddings and attention mechanisms into a progressive hierarchical extraction network to achieve domain-adaptive domain-related knowledge extraction, and it utilizes Stepwise Pivot Transformer networks and adaptive instance normalization to effectively utilize information from different modalities and domains. The experimental results have demonstrated the effectiveness of the proposed method.

However, the manuscript has suffered from the following problems.
1. Equation 2 may have a typo error, where there are two f-CLIP-I.
2. There are two L (Equation 8 and Equation 25), one is domain classfication loss, the other is fake news classfication loss. So, what is the final loss? Does final loss contain both losses? what's the relationships among LDomain, LDPLE and Li, Lt and Lm.
3. MOE  and CLIP is used to enhance the model's performance. Other works like [1-2] also use CLIP to help for debunking fake news at cross-modal setting. But the manuscript did not mention  and compare these methods. So, it is not clear, what makes the method achieve the better performance.
[1] Multimodal Fake News Detection via CLIP-Guided Learning. Arxiv 2022.
[2] Multi-modal Fake News Detection on Social Media via Multi-grained Information Fusion. ICMR 23.

**Strengths:**

A  multi-modality and multi-domain fake news detection method is proposed. From the pespective of cross-modal and cross-domain, it is new.

**Limitations:**

CLIP itself has the ablity of cross-modal and cross-domain generalization. MOE has the ability of permance boosting. It is not clear, what make the method proposed supass the other methods.

**Suitability:**

3

---

### Meta-Review · Area_Chair_Tu1V · 2024-07-02

**Recommendation:** Accept (Poster)
**Confidence:** 5

**Metareview:**

This paper focuses on multi-domain multi-modal fake news detection, which exhibits a more complex scenario. To address the inter-domain and intra-domain challenges, a multi-modality and multi-domain fake news detection method MMDFND is proposed. All reviews provide positive final ratings and believe that this submission is above the acceptance threshold with minor revisions. I recommend the acceptance of this paper, but the authors are strongly required to revise this paper carefully according to the reviews. Specially, the claim about "the first XXX" should raise a very serious caution.